# Chronic Sulforaphane Administration Inhibits Resistance to the mTOR-Inhibitor Everolimus in Bladder Cancer Cells

**DOI:** 10.3390/ijms21114026

**Published:** 2020-06-04

**Authors:** Saira Justin, Jochen Rutz, Sebastian Maxeiner, Felix K.-H. Chun, Eva Juengel, Roman A. Blaheta

**Affiliations:** 1Department of Urology, Goethe-University, 60590 Frankfurt am Main, Germany; justinsaira@hotmail.com (S.J.); Jochen.Rutz@kgu.de (J.R.); Sebastian.Maxeiner@kgu.de (S.M.); Felix.Chun@kgu.de (F.K.-H.C.); Eva.Juengel@unimedizin-mainz.de (E.J.); 2Department of Urology and Pediatric Urology, University Medical Center Mainz, 55131 Mainz, Germany

**Keywords:** bladder cancer, sulforaphane, mTOR, everolimus, drug resistance, growth, proliferation

## Abstract

Progressive bladder cancer growth is associated with abnormal activation of the mammalian target of the rapamycin (mTOR) pathway, but treatment with an mTOR inhibitor has not been as effective as expected. Rather, resistance develops under chronic drug use, prompting many patients to lower their relapse risk by turning to natural, plant-derived products. The present study was designed to evaluate whether the natural compound, sulforaphane (SFN), combined with the mTOR inhibitor everolimus, could block the growth and proliferation of bladder cancer cells in the short- and long-term. The bladder cancer cell lines RT112, UMUC3, and TCCSUP were exposed short- (24 h) or long-term (8 weeks) to everolimus (0.5 nM) or SFN (2.5 µM) alone or in combination. Cell growth, proliferation, apoptosis, cell cycle progression, and cell cycle regulating proteins were evaluated. siRNA blockade was used to investigate the functional impact of the proteins. Short-term application of SFN and/or everolimus resulted in significant tumor growth suppression, with additive inhibition on clonogenic tumor growth. Long-term everolimus treatment resulted in resistance development characterized by continued growth, and was associated with elevated Akt-mTOR signaling and cyclin-dependent kinase (CDK)1 phosphorylation and down-regulation of p19 and p27. In contrast, SFN alone or SFN+everolimus reduced cell growth and proliferation. Akt and Rictor signaling remained low, and p19 and p27 expressions were high under combined drug treatment. Long-term exposure to SFN+everolimus also induced acetylation of the H3 and H4 histones. Phosphorylation of CDK1 was diminished, whereby down-regulation of CDK1 and its binding partner, Cyclin B, inhibited tumor growth. In conclusion, the addition of SFN to the long-term everolimus application inhibits resistance development in bladder cancer cells in vitro. Therefore, sulforaphane may hold potential for treating bladder carcinoma in patients with resistance to an mTOR inhibitor.

## 1. Introduction

Bladder cancer is the ninth most common malignant ailment and the fourteenth most common cause of cancer death worldwide. Once metastasized, the disease is difficult to treat. Median survival with the MVAC (methotrexate, vinblastine, doxorubicin, cisplatin) or GC (gemcitabine, cisplatin) treatment plan is limited to 14.0 (GC) or 15.2 months (MVAC), with a five-year overall survival rate of 13.0% and 15.3%, respectively [1]. 

The PI3K (phosphoinositide-3 kinase) Akt (serine/threonine kinase)-mTOR (mammalian target of rapamycin) pathway is crucial for regulating cellular growth, proliferation, survival, and motility. In up to 40% of bladder cancers, mTOR pathway activation is closely involved with tumor progression [2]. Therefore, targeting the Akt-mTOR pathway is an attractive strategy to treat advanced urothelial carcinoma, and several mTOR inhibitors are clinically applied. However, treatment of bladder cancer with these inhibitors has not been as effective as expected. The potential benefit of the mTOR-inhibitors, temsirolimus and everolimus, has been demonstrated only for a subset of bladder cancer patients, and use of these drugs has been associated with severe, not well-tolerated, adverse events [3,4].

The lack of efficacy of mTOR targeting has been ascribed to the development of tumor cell resistance, forcing the activation of compensatory signaling pathways that dampen the potential of mTOR inhibitors [5]. For this reason, novel approaches are required to optimize cancer treatment.

Due to dissatisfaction with conventional treatment, many cancer patients turn to “alternative” or “complementary” (CAM) care options, hoping to lower their risk of cancer relapse and to actively contribute to their tumor therapy. Of the plentiful CAM options, oral use of natural herbs is common. Consumption of sulforaphane (SFN), an isothiocyanate rich in cruciferous vegetables such as broccoli, has been shown to exert chemopreventive properties, making it an interesting candidate for supplemental cancer treatment [6]. SFN acts as a natural histone deacetylase (HDAC) inhibitor with anti-tumor properties on prostate, lung, breast, and colon cancer [7]. There is also evidence that SFN not only inhibits bladder cancer growth, both in vitro and in vivo [8], but may also decrease resistance development and toxicity of conventional drug treatment [9]. Based on studies revealing that HDAC-inhibitors [10] with SFN in particular counteract everolimus resistance in renal cancer cell lines [11], it was postulated that SFN might be an innovative candidate to complement mTOR inhibition in bladder cancer therapy. Therefore, the purpose of this study was to evaluate whether SFN influences the growth and proliferation of bladder cancer cells in combination with everolimus under short- and long-term treatment, in vitro.

## 2. Results

### 2.1. Dose-Response-Analysis

A dose response analysis was carried out for everolimus, SFN, and combined everolimus+SFN. Everolimus, at 5, 1, or 0.5 nM, alone significantly reduced the number of RT112, UMUC3, and TCCSUP cells, independent of the concentration (Figure 1). SFN demonstrated distinct growth blockage at both 5 and 2.5 µM. Combined everolimus+SFN was not superior to the single drug treatment. At the low concentration setting (0.5 nM everolimus + 2.5 µM SFN), a trend towards a higher efficacy of the drug combination was observed in RT112 and TCCSUP cells. Apoptosis was also evaluated with the 5, 1, and 0.5 nM everolimus concentrations, each paired with 2.5 µM of SFN. SFN, everolimus, or the SFN-everolimus combination did not evoke signs of apoptosis, independent of the dosage used. Based on pilot studies, a drug dosage of 0.5 nM everolimus and 2.5 µM for SFN was set for further experiments.

### 2.2. Tumor Cell Proliferation under Short-Term Application

To evaluate the capacity of single tumor cells to grow into colonies (treated versus non-treated), a clonogenic assay was performed. The number of RT112 and TCCSUP clones was significantly diminished by everolimus or SFN, with the drug combination being more effective than each drug alone (Figure 2A). UMUC3 did not form clones and was therefore not evaluated. The BrdU incorporation assay reflected no difference in incorporation rate between everolimus-treated and control cells (all cell lines). SFN alone elevated BrdU in RT112 and UMUC3 but not in TCCSUP cells (Figure 2B). A further increase was seen when RT112 cells were treated with everolimus+SFN, whereas the response of UMUC3 cells to applying both compounds was similar to that for the SFN application. In TCCSUP, a significant reduction in the BrdU incorporation only became evident with the drug combination. 

### 2.3. Cell Cycling under Short-Term Treatment

To explore cell cycling, all cell lines were synchronized using aphidicolin. Following everolimus exposure, the number of G0/G1-phase tumor cells (all cell lines) increased, with a simultaneous decrease in S-phase (RT112) or G2/M-phase cells (UMUC3, TCCSUP), compared with the controls (Figure 3). In contrast, SFN evoked a considerable elevation of S-phase cells, along with a reduction in G0/G1- and G2/M-phase cells (all cell lines). The combined drug application was associated with an increased number of RT112 S-phase cells, and the effect was stronger compared with SFN alone. Elevation of S-phase cells was also seen in UMUC3 but not in TCCSUP cells, whereas G2/M-phase cells were down-regulated in all three cell lines in the presence of SFN+everolimus.

### 2.4. Cell Cycle Protein Profiling under Short-Term Treatment

Since initial studies showed a strong response of RT112 to SFN with a trend towards an additive response caused by SFN+everolimus (Figure 1), the cell cycle regulating proteins in synchronized RT112 cells were investigated. Figure 4 depicts proteins of the CDK-Cyclin-axis and Akt; Figure 5 is the mTOR submembers, Rictor and Raptor, histone H3 and H4 acetylation, as well as p19 and p27. Everolimus caused down-regulation of pAkt, CDK1, CDK2 (both total and phosphorylated (“p”)), and Cyclin A and B. SFN only diminished pCDK1 and CDK2, along with Cyclin A and B. Akt was elevated but pAkt was reduced by SFN. The effect of the everolimus+SFN application was different from the monotherapy in as much as pCDK1 was elevated, compared with the control. Similar to SFN alone, everolimus+SFN elevated Akt, but reduced pAkt and Cyclin A, compared with the control.

Raptor was lost in the presence of SFN or everolimus, whereas pRaptor was only significantly diminished by everolimus. pRictor was suppressed by all three drug regimens. Remarkably, aH3 was diminished by everolimus but increased following the SFN or everolimus+SFN applications. Acetylated H4 was not detectable in the controls and the everolimus-treated RT112 cells. However, aH4 was detected in the tumor cells treated with SFN or everolimus+SFN. Although p19 increased in all tumor cells subjected to SFN, everolimus, or everolimus+SFN, the strongest effects became evident under combined drug use. p27 strongly increased in cells treated with everolimus or everolimus+SFN, but not with SFN alone, where p27 was only slightly reduced.

### 2.5. Cell Growth and Proliferation under Long-Term Drug Application

Tumor cell behavior under chronic everolimus application was considerably different from the short-term application. The cell number was only marginally reduced (RT112) or not diminished at all (UMUC3, TCCSUP) when compared with the growth of untreated cells (Figure 6). Still, all tumor cell lines responded well to SFN or, even more strongly, to the SFN-everolimus drug combination.

Similar effects were seen with the clonogenic growth assay, where the long-term everolimus application did not alter the number of tumor cell clones, whereas SFN and SFN+everolimus did (drug combination > SFN alone) (Figure 7A). Proliferation of RT112 evaluated by the BrdU incorporation assay was even enhanced with the long-term everolimus application (Figure 7B). Proliferation was not altered (UMUC3) or only slightly diminished (TCCSUP) by everolimus, compared with the control cells. In contrast, SFN or SFN+everolimus (SFN+everolimus > SFN) still blocked clonogenic growth of RT112 and TCCSUP and prevented proliferation of UMUC3 (SFN) and TCCSUP (SFN+everolimus) (Figure 7A,B).

### 2.6. Cell Cycling under Long-Term Drug Treatment

Chronic everolimus application to RT112 led to an increase in both the G2/M- and S-phases, accompanied by a reduced number of G0/G1-phase cells (Figure 8), contrasting with the behavior in the short-term application model. SFN still induced similar effects on cell cycling as seen under the short-term application, i.e., loss of G0/G1- and elevation of G2/M- and S-phase cells. SFN+everolimus caused modifications similar to the everolimus monotherapy. In UMUC3 cells, there were no clear differences between the short- and long-term everolimus applications. However, the reduction in G2/M-phase cells in the long-term regimen was not as strong as was evident under the short-term treatment. SFN did not alter the number of S-phase UMUC3 cells, but reduced the G2/M- and up-regulated G0/G1-phase cells. The cell-phase distribution detected in the presence of everolimus and SFN+everolimus was similar. TCCSUP cells responded to the long-term everolimus application by accumulating in the G2/M and G0/G1 phases. The same response was also seen under the long-term SFN application, whereas the combination particularly elevated the number of G0/G1- but not G2/M-phase cells. TCCSUP S-phase cells were diminished independent of the kind of drug application.

### 2.7. Cell Cycle Protein Profiling under Long Term Drug Treatment

Figure 9 and Figure 10 are both related to synchronous RT112 cells and depict proteins of the CDK-Cyclin-axis and Akt (Figure 9), or proteins of the mTOR submembers, Rictor and Raptor, histone H3 and H4 acetylation as well as the proteins p19 and p27 (Figure 10). In contrast with the short-term application, everolimus caused a distinct up-regulation of pCDK1, pAkt, pRaptor, and pRictor. p19 and p27 were diminished following chronic everolimus exposure, and Cyclin A and H3 acetylation remained low.

Long-term SFN induced the same effects as observed under the short-term application with respect to most proteins analyzed: pCDK1, CDK2, Cyclin A, Cyclin B, Raptor, pRictor, p27 (all down-regulated compared with the controls); aH3 and aH4 (up-regulated compared with the controls). However, differences to the short-term application were seen with a few proteins. pCDK2 was reduced and pAkt and pRaptor were increased by SFN.

pAkt remained low in the presence of the SFN+everolimus combination, and pRaptor did not increase as was the case with SFN alone. Histone acetylation and the p19 expression level remained high, Cyclin A and B remained low with SFN+everolimus. In contrast with the early events, an increase in CDK1 as well as Akt and a p27 reduction were apparent under the long-term SFN+everolimus application.

### 2.8. Cell Cycle Protein Knock-Down

The Cyclin-CDK axis was profoundly altered by the compounds, with differences seen between everolimus and SFN and between the short- and long-term applications. The physiologic relevance of these particular proteins was therefore evaluated by focusing on CDK1-Cyclin B and CDK2-Cyclin A. Since p19 was modulated by everolimus and SFN in an additive manner, p19 knock-down studies were done as well. The efficacy of protein knock-down is depicted in Figure 11A–C. Loss of Cyclin A, Cyclin B, CDK1, or CDK2 was each associated with a significant cell growth stop in RT112 cells (72 h value). In contrast, p19 knock-down up-regulated cell growth.

## 3. Discussion

The data presented here demonstrate that the natural HDAC inhibitor, SFN, delays resistance induction caused by chronic everolimus exposure in a bladder cancer cell model. Clonogenic growth of RT112 and TCCSUP cells was better blocked in the short-term with everolimus+SFN than with either drug alone. Previous studies done on kidney carcinoma cells, using 1 nM everolimus paired with 5 nM SFN, have also shown additive blocking effects on tumor growth and proliferation [10]. The relevance of dually targeting mTOR and HDAC to prevent mitotic progression has also been demonstrated on other tumor entities such as prostate [12], ovarian, and mammary carcinoma [13]. Synergistic effects have been evoked on a panel of more than 60 human cancer cell lines treated with an mTOR-HDAC inhibitor combination, compared with single agent applications [14]. It is important to note that 5 µM SFN did not induce cytotoxic effects in the present investigation, nor has toxicity occurred in renal cell cancer cells exposed to 20 µM SFN [15]. The safety and tolerance of isothiocyanates in both healthy volunteers and cancer patients has been demonstrated [16,17]. SFN may, therefore, be a useful supplement to anti-cancer protocols based on down-regulating the Akt-mTOR signaling cascade.

Under the short-term treatment, everolimus or SFN slowed tumor cell growth equally well, though their underlying modes of action differed. Cell cycle analysis demonstrated an elevation of G0/G1 cells caused by everolimus, whereas SFN induced an elevation of S-phase cells. These differences are reflected in the cell cycle regulating protein profile. Everolimus strongly acted on pAkt, pRictor, pRaptor, pCDK1-Cyclin B, and pCDK2-Cyclin A, which were all down-regulated. In contrast, SFN down-regulated pCDK1-Cyclin B but neither pCDK2 nor pRaptor followed suit. Notably, acetylation of both H3 and H4 increased under SFN. It has recently been documented that HDAC may cross-communicate with the cell cycling machinery, so that H3 and H4 acetylation may trigger the loss of members of the Cyclin and CDK family [18]. Hypothetically, this mechanism may have played a role in the present investigation, and Cyclin B and CDK1 down-regulation might be considered a consequence of the increased histone acetylation induced by SFN.

Due to the different molecular mechanisms of SFN and everolimus, it is not surprising that combining them did not lead to additive effects in regard to the majority of analyzed proteins. Only the up-regulation of p19 and down-regulation of pRictor seen under the short-term treatment reflect an additive effect caused by the simultaneous drug application. Both proteins have been reported to be essential drivers of urothelial tumorigenesis and bladder cancer progression [19,20]. Hence, the additive response in the clonogenic growth and cell cycle assay might (at least partially) be attributed to altering the p19 and pRictor expression levels. Most proteins reacted similarly with the combined everolimus+SFN and SFN applications (except for p27, whose expression of which was similar with the combined and everolimus applications), indicating that SFN-triggered processes dominate those induced by everolimus. This is particularly important in view of the increased histone acetylation seen with SFN and the SFN-everolimus combination. Several studies indicate a close relationship between decreased histone acetylation and malignant bladder cancer progression [21,22]. Consequently, adding SFN to an everolimus-based treatment may amplify its anti-tumor efficacy by inhibiting not only mTOR but HDAC, as well. Based on a cell culture model, the HDAC inhibitor valproic acid has recently been shown to combat temsirolimus-resistant bladder cancer cells by increasing aH3 and aH4 [23].

The long-term everolimus application was associated with resistance induction. The tumor cell number was only marginally reduced (RT112) or not diminished (UMUC3, TCCSUP), and clonogenic growth was not altered. Cell cycle analysis revealed an increase in S- and/or G2/M-phase cells, paralleled by an increase in pCDK1, pAkt pRictor and pRaptor, and a decrease in p19 and p27. Therefore, chronic exposure to everolimus could create negative feedback loops with reactivation of Akt-mTOR signaling, driving the mitotic circle forward and blocking those proteins with tumor-suppressive properties (p19, p27). In this context, the p27 as well as the CDK1 expressions and functions have been demonstrated to depend on Akt-mTOR-signaling [24,25]. The same might be true for p19, whose expression has been associated with Akt hypophosphorylation [26]. Still, the relevance of p27 as a biomarker for bladder cancer remains controversial. Low p27 expression has been correlated with an increased tumor stage and shorter overall survival by some investigators [27,28], whereas others see no predictive value of p27 [29,30], or even observe high p27 associated with poor prognosis [31]. Increased p27 has also been associated with loss of bladder cancer cell proliferation in several in vitro experiments [32], whereas others point to a role of p27 in activating epithelial-mesenchymal transformation [33] and forcing tumor cell migration and invasion [34]. p27 was only moderately diminished by SFN and the SFN-everolimus combination. Therefore, the treatment regimen presented here prevents strong down-regulation induced by chronic everolimus alone. However, whether this action is clinically important cannot yet be definitively answered.

It is relevant that pCDK1, pCDK2, and pRictor remained low and p19 high under the long-term application of SFN. This molecular action may explain why chronic SFN application did not induce resistance, but rather maintained the cell growth inhibition that was established under the short-term SFN application. In good accordance, it has recently been shown that the long-term application of SFN does not induce resistance in renal or pancreatic cancer cells [35,36]. Still, attention should be paid to pAkt along with its down-stream target pRaptor, both of which were up-regulated by SFN after long-term exposure. Long-term administration of the HDAC inhibitors panobinostat or valproic acid has been shown to induce resistance triggered by Akt reactivation [37,38]. Speculatively, an elevation of Akt phosphorylation in the presence of SFN indicates early signs of resistance. In this context, clonogenic growth of RT112 cells following eight weeks SFN exposure was diminished to a lesser extent than under short-term SFN exposure. This requires further investigation, and unspecific side effects should not be excluded.

It is clinically important that combining SFN with everolimus potently blocked tumor growth and proliferation under the long-term application, better than the SFN application alone. Evidence has been presented showing that SFN reverses chemo-resistance to lapatinib, temozolomid, and cisplatin [39,40,41], opening a perspective that a combinatorial strategy with SFN could also inhibit resistance to everolimus. The SFN+everolimus application not only compensated for the p19 loss that occurred when everolimus was applied alone, but also induced a stronger p19 increase, compared with the SFN monotherapy. p19 is considered to be a “compensatory tumor barrier” in bladder cancer [42]. An increase in p19 has been shown to correlate with a marked increase in apoptotic bladder cancer cells [43] and, vice versa, a deletion of p19 has been shown to correlate with clinicopathological parameters associated with a worse prognosis [44]. We, therefore, assume that down-regulation of p19 caused by everolimus might be one parameter responsible for resistance induction. Hence, up-regulation of p19 seen in the presence of both everolimus and SFN is interpreted as a counter-regulatory process, caused by SFN. Since p19 is expressed to a higher extent by combining the drugs than by SFN alone, the intriguing question arises as to whether SFN may even re-induce chemosensitivity in bladder cancer cells.

Recently, a p19 increase has been traced back to HDAC inhibition. Pharmacological or transcriptional blockade of HDAC has been shown to trigger an elevated p19 expression, along with cell growth arrest [45]. This is of interest, since the aH3 expression level decreased under long-term everolimus but was up-regulated by SFN+everolimus. aH4 was also detectable in the presence of SFN+everolimus but not in the presence of everolimus alone. Overexpression of HDAC with low histone acetylation has been shown to promote growth and chemoresistance in urothelial cancer [46]. Therefore, enhanced histone acetylation evoked by SFN might be a further mechanism by which SFN combats resistance. However, whether histone acetylation triggers the p19 expression in our cell culture model has not been evaluated and, therefore, remains speculative. Whether or not direct cross-communication exists between aH3/aH4 and p19, HDAC inhibitors should be considered relevant in treating patients with solid tumors [47]. Hence, combining the mTOR inhibitor everolimus with a naturally derived epigenetic drug such as SFN may provide an innovative strategy in treating bladder cancer by acetylating histones H3 and H4 and concomitantly elevating the tumor suppressor p19. Though both CDK1 and Akt were up-regulated by SFN plus everolimus, indicating resistance buildup, phosphorylation of CDK1 remained low and the phosphorylated form of Akt was strongly diminished. Therefore, it may be assumed that the overall activity of CDK1 and Akt has been suppressed by the SFN-everolimus combination. This may also explain the superiority of the combination regimen over SFN single drug use.

Some ambivalence remains. pRaptor was only moderately elevated under the long-term everolimus therapy but was distinctly enhanced by SFN and SFN+everolimus. Whether pRaptor up-regulation influences tumor cell proliferation is not clear. HDAC inhibition has been reported to be associated with mTORC2 (Rictor) suppression [48], which may create a negative feedback loop, leading to an up-regulation of mTORC1 (Raptor) [49]. Ongoing studies are required to explore the relevance of a pRaptor increase to resistance acquisition.

Future work should be directed towards studies with animal models to verify in vitro findings and to optimize SFN drug dosage. Clinical trials including those with bladder cancer patients, which so far have not been done, should be performed to improve SFN’s bioavailability. It is encouraging to note that daily oral administration of 200 µM SFN resulted in a plasma level of 655 ng/mL that was well tolerated by melanoma patients [50]. The weekly consumption of a 300 mL portion of broccoli soup was also safe for patients with localized prostate cancer and, an inverse association between the intake of cruciferous vegetables and cancer progression was observed [51]. Positive results have also been reported in patients with pancreatic ductal adenocarcinoma under palliative chemotherapy, additionally treated with 15 capsules containing 508 μM sulforaphane and 411 μM glucoraphanin, administered daily for up to 1 year. However, although an improved outcome was noted, taking 15 capsules daily was difficult for some patients, and the broccoli sprouts sometimes increased digestive problems including nausea and emesis [52]. This problem necessitates the development of highly effective new sulforaphane formulations that are well tolerated.

## 4. Materials and Methods

### 4.1. Cell Cultures

Bladder carcinoma cell lines RT112 (pathological stage T2, moderately differentiated, grade 2/3), UMUC3 (high grade 3, invasive) (both: ATCC/LGC Promochem GmbH, Wesel, Germany) and TCCSUP (transitional cell carcinoma, grade 4) (DSMZ, Braunschweig, Germany) were cultured in RPMI 1640 medium (Gibco/Invitrogen, Karlsruhe, Germany) supplemented with 10% fetal bovine serum (FBS), 2% HEPES buffer, 1% GlutaMAX, and 1% penicillin/streptomycin (all: Gibco/Invitrogen), and incubated at 37 °C in a humidified incubator with 5% CO2.

### 4.2. Drug Application

Everolimus (Novartis Pharma AG, Basel, Switzerland) was dissolved in DMSO as a 10 mM stock solution and stored as aliquots at −20 °C. Prior to experiments, everolimus was diluted in the cell culture medium. L-sulforaphane was provided by Biomol, Hamburg, Germany (CAS registry number: CAS 142825-10-3). The optimal final drug concentrations were determined using dose response analyses with 5, 1, or 0.5 nM everolimus and 5 or 2.5 µM SFN. Untreated tumor cells served as controls. Pilot experiments demonstrated that 8 weeks chronic everolimus exposure leads to resistance development, expressed by increased tumor growth and proliferation, compared with respective controls [11]. Therefore, bladder cancer cells were chronically treated with everolimus at a defined concentration (0.5, 1, or 5 nM, three times a week) for 8 weeks. In separate experiments, bladder cancer cells were chronically treated with SFN (5 or 2.5 µM, three times a week) for 8 weeks. To investigate the drug combination, both everolimus and SFN together were added to the cell cultures for 8 weeks. Control experiments consisted of incubation with cell culture medium alone for 8 weeks. Both non-treated and treated cell cultures were passaged as soon as confluency was attained. After 8 weeks, the cell cultures were immediately subjected to the assays listed below, whereby the drugs remained in the cell culture medium throughout the assay. For short-term drug exposure, tumor cells were treated for 24 h with SFN or everolimus alone, or with the SFN+everolimus combination, and then used for experiments. Controls received cell culture medium without drugs. Short- (24 h) versus long-term drug exposure (8 weeks) was then compared. Prior to the experiments, cell growth behavior of the tumor cells was checked by the MTT assay (see below). Cell passages with a strong response to everolimus after 24 h and a loss of response after 8 weeks were employed for the experiments.

### 4.3. Tumor Cell Growth and Proliferation

Cell growth was evaluated using the 3-(4,5-dimethylthiazol-2-yl)-2,5-diphenyltetrazolium bromide (MTT) dye reduction assay (Roche Diagnostics, Penzberg, Germany). For each cell line, 5000 cells were pipetted in triplicate to 96-well plates with drug or normal medium. After 24, 48, and 72 h, 10 µL MTT (0.5 mg/mL) was added for an additional 4 h. Cells were then lysed in solubilization buffer (10% SDS in 0.01 M HCl) overnight at 37 °C, 5% CO_2_. Absorbance at 550 nm was recorded with a microplate enzyme-linked immunosorbent assay (ELISA) reader. To correlate absorbance with cell number, a defined number of cells ranging from 2500 to 160,000 cells/well was added to the microtiter plates (in triplicate). After subtracting the background absorbance (cell culture medium alone), results were expressed as mean cell number. Proliferation of synchronous cells was quantified using a non-isotopic enzyme immunoassay. To synchronize the cells, they were treated with aphidicolin (1 µg/mL; Enzo Life Sciences, Lörrach, Germany) for 24 h. Afterwards, the aphidicolin added medium was removed and cells were incubated for another 24 h with the drug or normal medium. To carry out the proliferation assay, the BrdU (Bromodeoxyuridine) cell proliferation ELISA kit (Calbiochem/Merck Biosciences, Darmstadt, Germany) was used. A total of 5000 cells were pipetted in triplicate into 96-well plates and incubated with BrdU for 24 h. Cells were then fixed and immunolabelling was carried out according to the manufacturer’s instructions.

### 4.4. Clonogenic Growth Assay

Bladder cancer cells were transferred to 6-well plates at 500 cells per well. An amount of 4 mL medium with or without (control) drugs was added to the wells to compare the short- versus long-term applications. Depending on the cell line, plates were incubated at 37 °C for 1–2 weeks until colonies were of sufficient size for counting. Colonies were fixed with 1% glutaraldehyde for 10 min at room temperature and counted. Colonies containing ≥50 cells were counted using a Zeiss ID 03 light microscope (Zeiss AG, Oberkochen, Germany).

### 4.5. Apoptosis

The Annexin V-FITC Apoptosis Detection kit (BD Pharmingen, Heidelberg, Germany) was used to detect the percentage of vital, necrotic, and apoptotic cells. Tumor cells were washed twice with phosphate-buffered saline (PBS) and then incubated with 5 µL of Annexin V-FITC and 5 µL propidium iodide (PI) in the dark for 15 min at room temperature. From each sample, 10,000 events were analyzed using FACScalibur (BD Biosciences, Heidelberg, Germany). The CellQuest software (BD Biosciences) was used for the data acquisition. To evaluate toxic effects of everolimus and SFN, cell viability was determined by trypan blue (Gibco/Invitrogen, Darmstadt, Germany).

### 4.6. Cell Cycling

Cell cycle analysis was performed on sub-confluent cell cultures using CycleTest™ Plus DNA Reagent Kit (Becton Dickinson). The test was carried out with synchronous cells. DNA was stained with PI according to the manufacturer’s instructions. Tumor cells were then subjected to flow cytometry using FACScan. For each sample, 10,000 events were observed. Data acquisition was carried out using the CellQuest software, and the ModFit software (Becton-Dickinson) was used to calculate the cell cycle distribution. The number of gated cells in the G1, G2/M, or S phases was expressed in percentage form.

### 4.7. Western Blot Analysis

Cell cycle regulating protein levels were examined in synchronized RT112 cells. Protein lysates were separated on 7%–12% polyacrylamide gel via electrophoresis at 100 V for 90 min. Proteins were then transferred to nitrocellulose membranes. Membranes were blocked with skim milk powder for 1 h and then incubated overnight with monoclonal antibodies directed against the following cell cycle proteins: CDK1 (IgG1, clone 1), pCDK1/Cdc2 (pY15; IgG1, clone 44/Cdk1/Cdc2), CDK2 (IgG2a, clone 55), Cyclin A (IgG1, clone 25), Cyclin B (IgG1, clone 18), p19 (IgG1, clone 52), Kip1/p27 (IgG1, clone 57; all from BD Biosciences), and pCDK2 (Thr160; Cell Signaling). The mTOR pathway was investigated through the following proteins: Raptor (clone 24C12), pRaptor (IgG, Ser792), Rictor (IgG, clone D16H9), pRictor (IgG, Thr1135, clone D30A3; all from Cell Signaling), PKBα/Akt (IgG1, clone 55), and pAkt (IgG1, pS472/pS473, clone 104A282; both from BD Biosciences). To investigate histone acetylation, marking was done with anti-histone H3 (IgG, clone Y173), anti-acetylated H3 (IgG, clone Y28), anti-histone H4 (polyclonal IgG), and anti-acetylated H4 (Lys8, polyclonal IgG; all from Cell Signaling). HRP-conjugated goat anti-mouse IgG and HRP-conjugated goat anti-rabbit IgG (both: Cell Signaling) served as the secondary antibody. Membranes were briefly incubated with ECL detection reagent (Amersham/GE Healthcare, München, Germany) to visualize the proteins and then analyzed by the Fusion FX7 (Peqlab, Erlangen, Germany). β-actin (Cell Signaling) served as the internal control. The GIMP 2.8 software was used to analyze the pixel density of the protein bands (both total and phosphorylated) and to calculate the ratio of protein intensity/β-actin intensity.

### 4.8. Blocking Studies

Transfection with small interfering RNA (siRNA) was carried out directed against CDK1 (gene ID: 983, target sequence: AAGGGGTTCCTAGTACTGCAA), cyclin B (gene ID: 891, target sequence: AATGTAGTCATGGTAAATCAA), cdk2 (gene ID: 1017, target sequence: AGGTGGTGGCGCTTAAGAAAA), cyclin A (gene ID: 890, target sequence: GCCAGCTGTCAGGATAATAAA), or p19 (gene ID: 1032, target sequence: ACCCAAGGCAGAGCATTTAA9; all: Qiagen, Hilden, Germany). Then, 3 × 105 cells were incubated with the transfection solution of siRNA and transfection reagent (HiPerFect Transfection Reagent; Qiagen) at a ratio of 1:6. Non-treated cells and cells treated with the 5 nM control siRNA (AllStars Negative Control siRNA; Qiagen) served as controls. Afterwards, the protein expression level and tumor cell growth were analyzed as described above.

### 4.9. Statistics

The mean +/− SD was calculated. To exclude coincidence, all experiments were repeated three to five times. Statistical significance was evaluated with the “Wilcoxon–Mann–Whitney-U-Test” and “Student’s *T*-Test”. A *p*-value of 0.05 or less was considered significant.

## 5. Conclusions

The present findings reveal that chronic use of the mTOR inhibitor everolimus is associated with resistance development in bladder cancer cells, resulting in aggressive regrowth and tumor progression. Pairing everolimus with the natural HDAC inhibitor sulforaphane inhibited resistance induction caused by everolimus. Patients with bladder cell carcinoma may, therefore, benefit from an anti-tumor strategy including sulforaphane as a complementary component to everolimus.

## Figures and Tables

**Figure 1 ijms-21-04026-f001:**
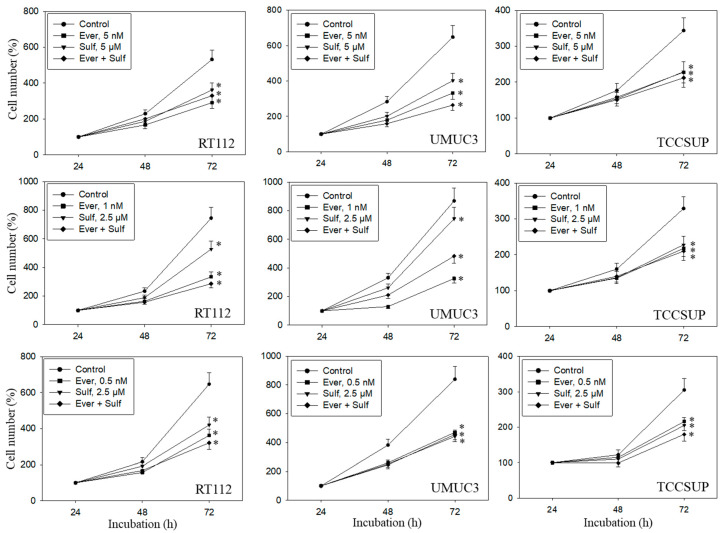
Cell number in response to everolimus (Ever- 5, 1, or 0.5 nM), sulforaphane (Sulf- 5, 2.5 µM), or (Ever + Sulf) under short-term application in bladder cancer cell lines RT112, UMUC3, and TCCSUP. Untreated cells served as controls. Cell number was evaluated after 24 (100%), 48, and 72 h by the MTT assay. Bars indicate standard deviation. Experiments were repeated five times. * indicates significant difference to untreated controls, *p* ≤ 0.05.

**Figure 2 ijms-21-04026-f002:**
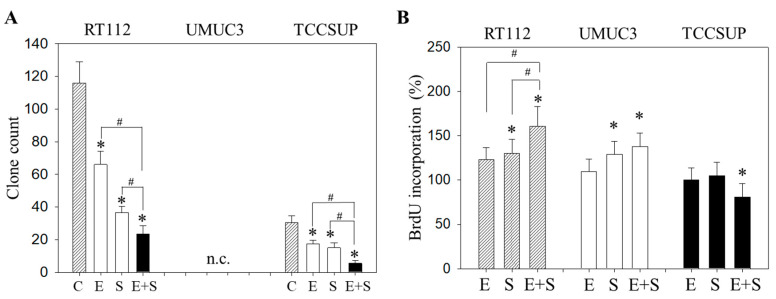
Evaluation of clonogenic growth (**A**) and BrdU incorporation (**B**) under short-term application of 0.5 nM everolimus (E) or 2.5 µM sulforaphane (S) or 0.5 nM everolimus + 2.5 µM sulforaphane (E + S). Control cells (C) remained untreated. RT112 clones were counted at day 8 and TCCSUP at day 10 following incubation. UMUC3 cells did not form clones (n.c.- not counted). The BrdU assay was carried out with synchronized cells with untreated control cells set at 100%. * indicates significant difference to untreated controls. # indicates significant difference between the mono and the combined applications.

**Figure 3 ijms-21-04026-f003:**
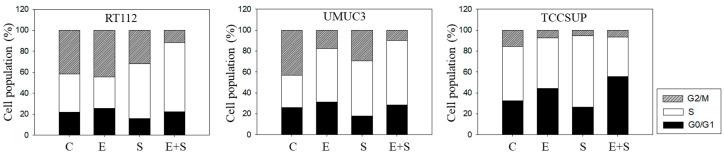
Cell cycle analysis—short-term treatment of synchronized cells with 0.5 nM everolimus (E) or 2.5 µM sulforaphane (S) or 0.5 nM everolimus+2.5 µM sulforaphane (E + S). Untreated cells served as controls (C). Percentage of RT112, UMUC3, or TCCSUP cells in G0/G1, S and G2/M-phase is indicated. Inter-assay variation <10%, intra-assay variation <40%.

**Figure 4 ijms-21-04026-f004:**
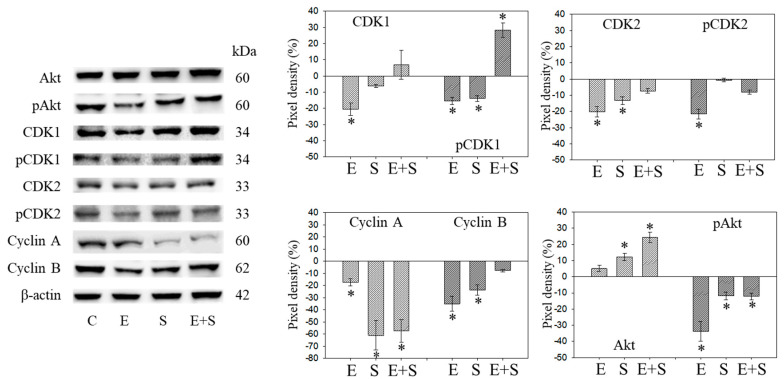
Protein profile of cell cycle regulating proteins (Akt, CDKs, Cyclins) after short-term exposure to 0.5 nM everolimus (E) or 2.5 µM sulforaphane (S) or 0.5 nM everolimus + 2.5 µM sulforaphane (E + S) in synchronized RT112 tumor cells. Controls (C) received cell culture medium alone. One representative of three separate experiments is shown. Each protein analysis was accompanied by a β-actin loading control. One representative internal control is shown here. The ratio of protein intensity/β-actin intensity was calculated and expressed as a percentage of the controls, set to 100%. * indicates significant difference to controls, *p* ≤ 0.05.

**Figure 5 ijms-21-04026-f005:**
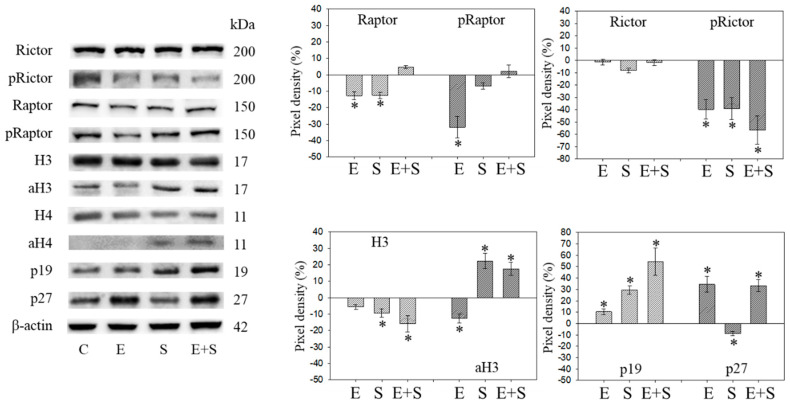
Protein profile of cell cycle regulating proteins (Rictor, Raptor, histone acetylation, p19, p27) after short-term exposure to 0.5 nM everolimus (E) or 2.5 µM sulforaphane (S) or 0.5 nM everolimus + 2.5 µM sulforaphane (E + S) in synchronized RT112 tumor cells. Controls (C) received cell culture medium alone. One representative of three separate experiments is shown. Each protein analysis was accompanied by a β-actin loading control. One representative internal control is shown. * indicates significant difference to controls, *p* ≤ 0.05.

**Figure 6 ijms-21-04026-f006:**
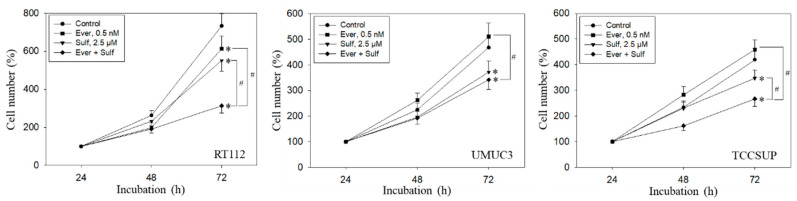
Growth of RT112, UMUC3, and TCCSUP cells exposed to 0.5 nM everolimus (Ever) or 2.5 µM sulforaphane (Sulf) or 0.5 nM everolimus + 2.5 µM sulforaphane (Ever + Sulf). Control cells (Control) remained unexposed. Cells were incubated in 96-well-plates for 24, 48, and 72 h. Cell number was set to 100% after 24 h incubation. Experiments were repeated five times. * indicates significant difference to untreated controls. # indicates significant difference between the mono and combined applications.

**Figure 7 ijms-21-04026-f007:**
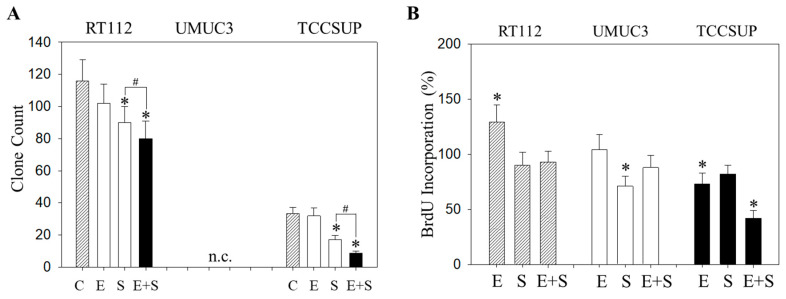
Evaluation of clonogenic growth (**A**) and BrdU incorporation (**B**) under the long-term application of 0.5 nM everolimus (E) or 2.5 µM sulforaphane (S) or 0.5 nM everolimus + 2.5 µM sulforaphane (E + S). Control cells (C) remained untreated. RT112 clones were counted at day 8 and TCCSUP at day 10 following incubation. UMUC3 cells did not form clones (n.c.- not counted). The BrdU assay was carried out with synchronized cells with untreated control cells set at 100%. * indicates significant difference to untreated controls. # indicates significant difference between the monotherapy and the combination therapy.

**Figure 8 ijms-21-04026-f008:**
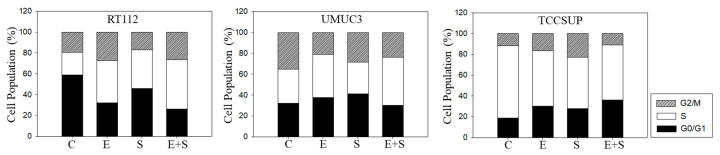
Cell cycle analysis—long-term treatment of synchronized RT112, UMUC3, and TCCSUP cells with 0.5 nM everolimus (E) or 2.5 µM sulforaphane (S) or 0.5 nM everolimus + 2.5 µM sulforaphane (E + S). Untreated cells served as the controls (C). Percentage of cells in the G0/G1-, S-, and G2/M-phase is indicated. Inter-assay variation <10%, intra-assay variation <40%.

**Figure 9 ijms-21-04026-f009:**
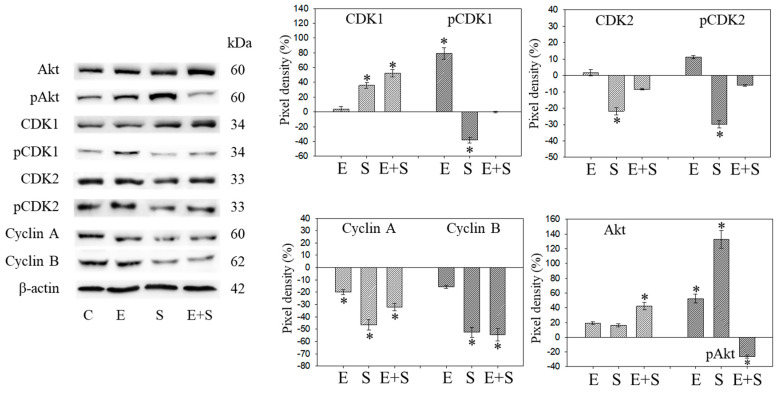
Protein profile of cell cycle regulating proteins (Akt, CDKs, Cyclins) after long-term exposure to 0.5 nM everolimus (E) or 2.5 µM sulforaphane (S) or 0.5 nM everolimus + 2.5 µM sulforaphane (E + S) to synchronized RT112 tumor cells. Controls (C) received cell culture medium alone. One representative of three separate experiments is shown. Each protein analysis was accompanied by a β-actin loading control. One representative internal control is shown. The right panel shows the results of the pixel density analysis. The ratio of protein intensity/β-actin intensity was calculated and expressed as a percentage of the controls, set to 100%. * indicates significant difference to controls, *p* ≤ 0.05.

**Figure 10 ijms-21-04026-f010:**
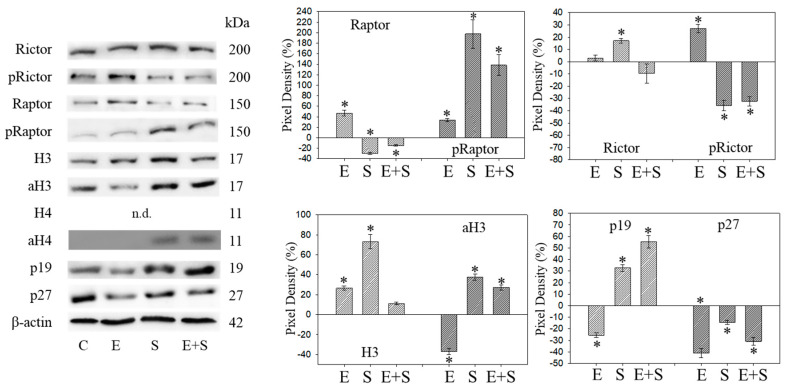
Protein profile of cell cycle regulating proteins (Rictor, Raptor, histone acetylation, p19, p27) after the long-term application of 0.5 nM everolimus (E) or 2.5 µM sulforaphane (S) or 0.5 nM everolimus + 2.5 µM sulforaphane (E + S) to synchronized RT112 tumor cells. Controls (C) received cell culture medium alone. One representative of three separate experiments is shown. Each protein analysis was accompanied by a β-actin loading control. One representative internal control is shown. The ratio of protein intensity/β-actin intensity is expressed as percentage of controls, set to 100%. * indicates significant difference to controls, *p* ≤ 0.05.

**Figure 11 ijms-21-04026-f011:**
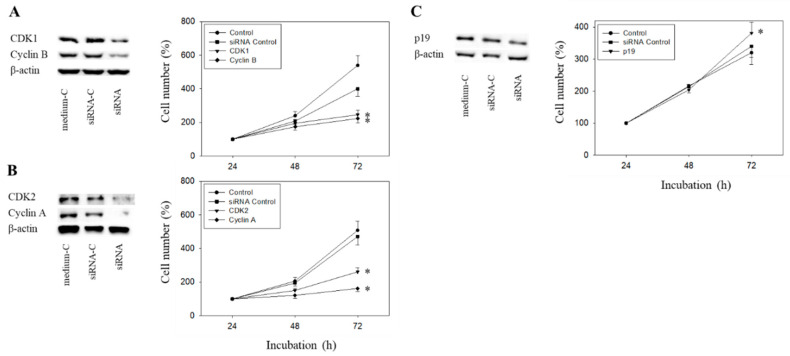
Tumor cell growth after functional blocking with small interfering RNA (siRNA) targeting CDK1/cyclin B (**A**), CDK2/Cyclin A (**B**), and p19 (**C**) of RT112 cells. Controls remained untreated. medium-C = cells treated with culture medium alone, siRNA-C = cells treated with unspecific siRNA, siRNA = cells treated with specific siRNA). β-actin served as the loading control. Protein data are shown on the left, growth response on the right side of the figure. One representative of three separate experiments is shown. * indicates significant difference to control.

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
