# Peer review of "Chronic Sulforaphane Administration Inhibits Resistance to the mTOR-Inhibitor Everolimus in Bladder Cancer Cells"

_ijms, 2020, doi:10.3390/ijms21114026_

Round 1
Reviewer 1 Report
This study indicated that chronic sulforaphane may inhibit the resistance effect of everolimus in bladder cancer cells. Author mentioned the long-term and short-term model have different effect after SFN treatment; however, how these model established has not been detailly described in material. In addition, the presentation of protein expression should be rechecked. Whether the quantification results of phosphorylation proteins have already been normalized by total protein and housekeeping gene?
- In the figure 1, I suggest to perform combination index (CI). It will be clearer to identify the best combination concentration.
- In western blotting (figure 4,5,9 and 10), the phosphorylation proteins should divide total form of proteins. What is the final value of phosphorylation proteins?
- In material and methods 4.3 tumor cell growth was validated by MTT assay, how author converse the absorbance into cell number should be addressed in material. Is it better to display as viability?
- The procedure of how drugs resistance model was established and maintained should be described in detail.
- To strengthen author’s proposal, we suggested that animal experiment should be performed.
- The role of p19 and p27 on bladder cancer should be mentioned in introduction.
Author Response
Comment 1: In the figure 1, I suggest to perform combination index (CI). It will be clearer to identify the best combination concentration.
Our answer: The pilot studies documented in figure 1 were based on earlier studies on renal cell carcinoma cells demonstrating significant anti-tumor effects of 2.5 and 5 µM sulforaphane and 1 and 5 nM everolimus (please see reference 11; reference 15). However, combining the drugs in the bladder cancer cell model did not lead to additive effects in the MTT-assay, independent of the concentration used (figure 1). We, therefore, decided to proceed with the “best” treatment strategy, i.e. the lowest concentrations of 0.5 nM (everolimus) and 2.5 µm (sulforaphane), both of which significantly suppressed tumor cell growth alone. We agree that further drug combinations could possibly be superior to the one presented here. However, the main focus of this investigation was to evaluate the effect of sulforaphane on everolimus-resistant tumor cells, thereby presenting a “realistic” treatment scenario. After all, consumption of sulforaphane (broccoli) should be kept to a reasonable amount. It is quite important that even 2.5 µM sulforaphane already evoked an anti-tumor response in the resistant cell lines. Please note that this concentration can be achieved in humans, indicating that the concentration used by us is clinically relevant (Shapiro et al. Safety, tolerance, and metabolism of broccoli sprout glucosinolates and isothiocyanates: a clinical phase I study. Nutr Cancer. 2006; 55: 53-62). Therefore, our in vitro results in principle do present evidence that sulforaphane might serve as an attractive anti-tumor compound. However, as we noted in the “Discussion”, studies with animal models are necessary to verify the in vitro findings, to optimize the drug dosage (e.g. by evaluating the combination index) and sulforaphane’s bioavailability.
Comment 2: In western blotting (figure 4,5,9 and 10), the phosphorylation proteins should divide total form of proteins. What is the final value of phosphorylation proteins?
Our answer: We quantified both the content of the total protein and the phosphorylated protein (all related to β-actin). This is now explained in more detail. Section 4.7. ‘Western blot analysis’ now reads: “β-actin (Cell Signaling) served as the internal control. GIMP 2.8 software was used to analyze pixel density of the protein bands (both total and phosphorylated) and to calculate the ratio of protein intensity/β-actin intensity”. We agree that it is important to compare alterations of a particular protein in its total and phosphorylated form. Still, it was not feasible to calculate the total/phosphorylated ratio of the pixel densities since each protein was evaluated by a different antibody with a different binding activity and quality. Furthermore, the employed antibody concentrations differed, as did the exposure time. These technical aspects do not allow direct comparison of the density of the different protein bands. Please note, for example, that the signal of pRaptor and pRictor is stronger than the one for the respective total protein in the control cells in figure 5. This may not reflect the real situation.
Nevertheless, the expression level of CDK1 versus pCDK1 and of Akt versus pAkt under long-term treatment with SFN and everolimus (figure 9) is of interest. We have taken this into consideration and included in the ‘Discussion’:
“Hence, combining the mTOR-inhibitor, everolimus, with a naturally derived epigenetic drug such as SFN may provide an innovative strategy in treating bladder cancer by acetylating histones H3 and H4 and concomitantly elevating the tumor suppressor p19. Though both CDK1 and Akt were up-regulated by SFN plus everolimus, indicating resistance buildup, phosphorylation of CDK1 remained low and the phosphorylated form of Akt was strongly diminished. Therefore, it may be assumed that the overall activity of CDK1 and Akt has been suppressed by the SFN-everolimus-combination. This may also explain the superiority of the combination regimen over SFN single drug use”.
Comment 3: In material and methods 4.3 tumor cell growth was validated by MTT assay, how author converse the absorbance into cell number should be addressed in material. Is it better to display as viability?
Our answer: The MTT-assay is now described in more detail: “Cells were then lysed in solubilization buffer (10% SDS in 0.01 M HCl) overnight at 37°C, 5% CO2. Absorbance at 550 nm was recorded with a microplate enzyme-linked immunosorbent assay (ELISA) reader. To correlate absorbance with cell number, a defined number of cells ranging from 2,500-160,000 cells/well were added to the microtiter plates (in triplicate). After subtracting background absorbance (cell culture medium alone), results were expressed as mean cell number”.
Comment 4: The procedure of how drugs resistance model was established and maintained should be described in detail.
Our answer: In the present investigation tumor cells were chronically treated over 8 weeks with everolimus at a constant dose of 0.5 nM in all experiments, excepting the pilot study where 0.5, 1, and 5 nM were employed. A respective reference has been included (reference 11). After 8 weeks, the tumor cells were immediately subjected to the assays listed in the methods section. To maintain similar assay conditions tumor cells incubated for more than 8 weeks were not used so that we did not evaluate tumor cell behavior treated with everolimus beyond 8 weeks. Please note that the concept described here differs from another resistance model applied in former investigations, where everolimus resistance was induced by treating tumor cells with stepwise ascending concentrations from 1 nM up to 1 μM for over one year and then maintained in culture medium enriched with 1 µM everolimus (E.g. Juengel et al. Mol Cancer 2014;13:152).
To make this point clear, we have expanded the respective methods part which now reads: “Pilot experiments demonstrated that 8 weeks chronic everolimus exposure leads to resistance development, expressed by increased tumor growth and proliferation, compared to respective controls [11]. Therefore, bladder cancer cells were chronically treated with everolimus at a defined concentration (0.5, 1, or 5 nM, three times a week) for 8 weeks. In separate experiments, bladder cancer cells were chronically treated with SFN (5 or 2.5 µM, three times a week) for 8 weeks. To investigate the drug combination, both everolimus and SFN together were added to the cell cultures for 8 weeks. Control experiments consisted of incubation with cell culture medium alone for 8 weeks. Both non-treated and treated cell cultures were passaged as soon as confluency was attained. After 8 weeks, the cell cultures were immediately subjected to the assays listed below, whereby the drugs remained in the cell culture medium throughout the assay. For short term drug exposure tumor cells were treated for 24 h with SFN or everolimus alone, or with the SFN+everolimus combination and then used for experiments. Controls received cell culture medium without drugs. Short- (24 h) versus long-term drug exposure (8 weeks) was then compared. Prior to the experiments, cell growth behavior of the tumor cells was checked by the MTT-assay (see below). Cell passages with a strong response to everolimus after 24 h and a loss of response after 8 weeks were employed for experiments”.
Comment 5: To strengthen author’s proposal, we suggested that animal experiment should be performed.
Our answer: We absolutely agree that in vivo studies must validate the in vitro data. However, establishing an adequate in vivo resistance model is quite complex and we have not as yet been successful in establishing one. Unfortunately, resistant tumor cells regain drug sensitivity a few weeks after the s.c. injection of everolimus-resistant tumor cells, although mice were continuously treated with everolimus thereafter. We, therefore, regret that we cannot provide the reader with valid in vivo data. To still consider the referee’s comment, we have included additional information about clinical trials with sulforaphane in the ‘Discussion’ that points to the importance of our results (please also see our response to comment 1 of referee 2):
“Future work should be directed towards studies with animal models to verify in vitro findings and to optimize SFN drug dosage. Clinical trials including those with bladder cancer patients, which so far have not been done, should be performed to improve SFN’s bioavailability. It is encouraging to note that daily oral administration of 200 µM SFN resulted in a plasma level of 655 ng/mL that was well tolerated by melanoma patients [50]. The weekly consumption of a 300 mL portion of broccoli soup was also safe for patients with localized prostate cancer and, an inverse association between the intake of cruciferous vegetables and cancer progression was observed [51]. Positive results have also been reported in patients with pancreatic ductal adenocarcinoma under palliative chemotherapy, additionally treated with 15 capsules containing 508 μM sulforaphane and 411 μM glucoraphanin, administered daily for up to 1 year. However, although improved outcome was noted, taking 15 capsules daily was difficult for some patients, and the broccoli sprouts sometimes increased digestive problems including nausea and emesis [52]. This problem necessitates the development of highly effective new sulforaphane formulations that are well tolerated.”
Comment 6: The role of p19 and p27 on bladder cancer should be mentioned in introduction.
Our answer: The relevance of p19 in bladder cancer has been dealt with in much detail in the ‘Discussion’. We, therefore, would prefer not to repeat this line of thought in the “Introduction” (although we are prepared to do this, if still required by the referee). The role of p27 in bladder cancer is controversially discussed. Low p27 expression has been correlated with an increased tumor stage and shorter overall survival by some authors (Sarsik et al. Pathol. Oncol. Res. 2016;22:839-845. Grapsa et al. J. BUON. 2014;19:1121-1124.), whereas others associated no predictive value with p27 (da Silva et al. Pathol. Oncol. Res. 2020;26:175‐181. Passoni et al. Urol. Oncol. 2016;34:485.e7-485.e14.). Some have even observed an inverse association between p27 expression and prognosis (Fahmy et al. Hum. Pathol. 2013;44:1766-1772.). Increase of p27 has also been associated with loss of bladder cancer cell proliferation in several in vitro experiments (Islam et al. Target Oncol. 2016;11:209-227.), whereas others point to a role of p27 in activating epithelial-mesenchymal transformation (Yoon et al. Proc Natl Acad Sci U S A. 2019;116:7005-7014) and forcing tumor cell migration and invasion (Park et al. Int J Oncol. 2014;44:1349-1356). We have now discussed the role of p27 in view of our data in the ‘Discussion’, which now reads:
“In this context, p27 as well as CDK1 expression and function has been demonstrated to depend on Akt-mTOR-signaling [24,25]. The same might be true for p19, expression of which has been associated with Akt hypophosphorylation [26]. Still, the relevance of p27 as a biomarker for bladder cancer remains controversial. Low p27 expression has been correlated with an increased tumor stage and shorter overall survival by some investigators [27,28], whereas others see no predictive value of p27 [29,30], or even observe high p27 associated with poor prognosis [31]. Increased p27 has also been associated with loss of bladder cancer cell proliferation in several in vitro experiments [32], whereas others point to a role of p27 in activating epithelial-mesenchymal transformation [33] and forcing tumor cell migration and invasion [34]. p27 was only moderately diminished by SFN and the SFN-everolimus-combination. Therefore, the treatment regimen presented here prevents strong down-regulation induced by chronic everolimus alone. However, whether this action is clinically important cannot yet be definitively answered”.
Reviewer 2 Report
The current manuscript, "Chronic sulforaphane administration inhibits resistance to the mTOR-inhibitor everolimus in bladder cancer cells", by Justin et al. is of significant interest to the urologic cancer field. The in vitro study is substantial and the conclusions are justified by the data.
My first comment on the study has to do with the understood problems/limitations with using long-established tumor cell lines. I'm wondering whether the authors considered using more newly-established primary patient-derived cell lines or even perhaps PDX mice? These additional studies may help to corroborate the observations.
I don't feel the study sufficiently addressed this idea of long-term culture and since long-term treatments are more relevant to clinical use, I feel that some more studies here are important. Have authors looked at long-term culture in the presence of drugs exceeding 10 days? Do the cells eventually die out? Are they capable of being passaged and what do their growth curves look like? Are the effects of the drugs transient? Do the cells return to full proliferative capacity when the drugs are removed? The reduction of clones is at 8 and 10 days is not huge, although significant. With only a single, "long" time-point and limited readouts of the fate of cells over time, I don't think the data gives us a feel for whether SFN would truly be an effective additive measure for everolimus. I feel that longer and more comprehensive studies on cell growth would strengthen this portion of the study and complement the high quality molecular biology analyses. At minimum, perhaps the authors could address some of these questions in the discussion section.
Author Response
Comment, part 1: My first comment on the study has to do with the understood problems/limitations with using long-established tumor cell lines. I'm wondering whether the authors considered using more newly-established primary patient-derived cell lines or even perhaps PDX mice? These additional studies may help to corroborate the observations.
Our answer: We agree that studies with patient-derived primary tumor cells may distinctly enhance the value of the in vitro data presented here. However, we were not successful in establishing drug-resistant tumor cells isolated from patient tissue. Most patient-derived tumor cells rapidly die a few days after starting cultivation. Even resistance induction in established cell lines is complex and not all cell lines lose drug-sensitivity over time. We, therefore, were pleased to be able to provide data from a panel of three cell lines. We are aware that the results should be seen in this context and may not be transferred to patients. Still, in vitro resistance induction by chronic everolimus exposure to the cell lines does reflect clinical reality. Concerning SFN our data point to the principal therapeutic relevance of this compound due to its efficacy even after 8 weeks application. To strengthen our message, we have now included some information about clinical trials with sulforaphane in the ‘Discussion’, which may corroborate our in vitro observation (please also see our response to comment 5 of referee 1):
“Future work should be directed towards studies with animal models to verify in vitro findings and to optimize SFN drug dosage. Clinical trials including those with bladder cancer patients, which so far have not been done, should be performed to improve SFN’s bioavailability. It is encouraging to note that daily oral administration of 200 µM SFN resulted in a plasma level of 655 ng/mL that was well tolerated by melanoma patients [50]. The weekly consumption of a 300 mL portion of broccoli soup was also safe for patients with localized prostate cancer and, an inverse association between the intake of cruciferous vegetables and cancer progression was observed [51]. Positive results have also been reported in patients with pancreatic ductal adenocarcinoma under palliative chemotherapy, additionally treated with 15 capsules containing 508 μM sulforaphane and 411 μM glucoraphanin, administered daily for up to 1 year. However, although improved outcome was noted, taking 15 capsules daily was difficult for some patients, and the broccoli sprouts sometimes increased digestive problems including nausea and emesis [52]. This problem necessitates the development of highly effective new sulforaphane formulations that are well tolerated.”.
Comment, part 2: I don't feel the study sufficiently addressed this idea of long-term culture and since long-term treatments are more relevant to clinical use, I feel that some more studies here are important. Have authors looked at long-term culture in the presence of drugs exceeding 10 days? Do the cells eventually die out? Are they capable of being passaged and what do their growth curves look like? Are the effects of the drugs transient? Do the cells return to full proliferative capacity when the drugs are removed?
Our answer: There are several strategies to induce resistance. In former studies, we investigated cardinal differences between drug resistant and sensitive tumor cells. Here, everolimus resistance was induced by treating tumor cells with stepwise ascending concentrations from 1 nM up to 1 μM for over one year and then maintaining the cells in culture medium enriched with 1 µM everolimus (E.g. Juengel et al. Mol Cancer 2014; 13: 152).
In contrast, the present investigation was designed to evaluate tumor cell behavior in the presence of a drug combination (versus single drug treatment). It was not the primary endpoint to compare drug-resistant versus drug-sensitive tumor cells. We were therefore interested in analyzing the tumor cell response to a drug regimen applied chronically over a defined time period. Tumor cells were treated chronically over 8 weeks with everolimus at a constant dosage. After 8 weeks, the tumor cells were immediately subjected to the assays listed in the methods section. Tumor cells incubated for more than 8 weeks were not used so as to maintain similar assay conditions. The 8 weeks protocol was chosen since earlier experiments demonstrated 8 weeks to be sufficient to evoke everolimus resistance. Indeed, extension to 12 weeks did not alter the tumor cell response to everolimus (E.g. Juengel et al. Cancer Lett. 2012; 324: 83-90).
During incubation, the tumor cells were passaged as soon as confluency was attained. The “chronic” cell cultures received everolimus three times a week. The “acute” cell cultures were incubated in cell culture medium alone and received everolimus prior to the experiments. The same was done with SFN. We also routinely checked tumor cell growth curves by the MTT-test. Only cells with a high growth capacity in the controls (no everolimus) and with differences between acute (strong response to everolimus) and chronic treatment (no response or loss of response to everolimus) were subjected to the assays.
Please note: Chronic treatment was for 8 weeks, not just for 10 days. The 10 day period is related to a standard protocol to count tumor cell clones. However, even in this case, the tumor cells were previously cultivated for 8 weeks and then exposed to the clonogenic growth assay.
We apologize that the respective methods have not been clearly described and caused confusion. The methods part now reads:
“Pilot experiments demonstrated that 8 weeks chronic everolimus exposure leads to resistance development, expressed by increased tumor growth and proliferation, compared to respective controls [11]. Therefore, bladder cancer cells were chronically treated with everolimus at a defined concentration (0.5, 1, or 5 nM, three times a week) for 8 weeks. In separate experiments, bladder cancer cells were chronically treated with SFN (5 or 2.5 µM, three times a week) for 8 weeks. To investigate the drug combination, both everolimus and SFN together were added to the cell cultures for 8 weeks. Control experiments consisted of incubation with cell culture medium alone for 8 weeks. Both non-treated and treated cell cultures were passaged as soon as confluency was attained. After 8 weeks, the cell cultures were immediately subjected to the assays listed below, whereby the drugs remained in the cell culture medium throughout the assay. For short term drug exposure tumor cells were treated for 24 h with SFN or everolimus alone, or with the SFN+everolimus combination and then used for experiments. Controls received cell culture medium without drugs. Short- (24 h) versus long-term drug exposure (8 weeks) was then compared. Prior to the experiments, cell growth behavior of the tumor cells was checked by the MTT-assay (see below). Cell passages with a strong response to everolimus after 24 h and a loss of response after 8 weeks were employed for experiments”.
We did not investigate whether tumor cells may lose resistance or whether the cells regain full proliferative capacity when the drugs have been removed. This was not the concern of the present investigation. However, we know from former studies that resistance may be temporary and tumor cells with initially established resistance may regain drug sensitivity when re-cultivated in drug-free medium. However, the time of reversal to drug-sensitivity strongly depends on the tumor type, the subline and drug used. The response of drug-sensitive tumor cells also varies. Depending on drug metabolism, the wash out phase, and bioavailability, several days after removing the drug from the cell culture medium tumor cells may regain full proliferative activity. Still, this was not the focus of the present investigation and the tumor cells were treated with SFN and/or everolimus throughout the experiments. This has now been clarified in “Methods”.
Comment, part 3: The reduction of clones is at 8 and 10 days is not huge, although significant. With only a single, "long" time-point and limited readouts of the fate of cells over time, I don't think the data gives us a feel for whether SFN would truly be an effective additive measure for everolimus. I feel that longer and more comprehensive studies on cell growth would strengthen this portion of the study and complement the high quality molecular biology analyses. At minimum, perhaps the authors could address some of these questions in the discussion section.
Our answer: Our experimental design was based on the main question of whether chronic treatment with SFN may counteract everolimus-caused resistance, evaluated in terms of cell growth and proliferation. We concentrated on the 8 week period, since we have documented earlier that this incubation time is sufficient to induce everolimus-resistance. To obtain a detailed overview of tumor cell growth characteristics, several assays were included: the MTT-test, counting the tumor cell number dynamically over 3 days, the BrdU-test analyzing the proliferation rate, which was accompanied by analyzing cell cycle progression and, finally, the growth of single tumor cells after 8-10 days. We, therefore, believe that we have satisfactorily dealt with tumor cell fate, at least over a period of 8 weeks. Earlier studies demonstrated no distinct differences of the tumor cell behavior when the incubation period was extended to 12 weeks.
Indeed, no massive reduction of RT112 clones under SFN or the SFN-everolimus combination was apparent. Two things may account for this. Either the SFN-concentration was too low (pointing to early signs of resistance development?) or the remaining clones were derived from cells initially resistant to drug treatment. It is correct that we cannot conclude anything about cell cultures being exposed to the drugs for more than 8 weeks. Although there is evidence that long-term application of SFN over months does not induce resistance in renal or pancreatic cancer cells (reference 27,28), this may not hold true with respect to bladder carcinoma. Certainly, important questions remain. Chronic consumption of SFN over months may also create negative feedback loops leading to a loss of activity (we have already pointed to the elevation of pAkt in this context in the “Discussion”). We have now further addressed this issue in the “Discussion”:
“Speculatively, elevation of Akt phosphorylation in the presence of SFN indicates early signs of resistance. In this context, clonogenic growth of RT112 cells following 8 weeks SFN-exposure was diminished to a lesser extent than under short-term SFN-exposure. This requires further investigation, and unspecific side effects should not be excluded”.